# Yolk Fatty Acid Content, Lipid Health Indices, and Oxidative Stability in Eggs of Slow-Growing Sasso Chickens Fed on Flaxseed Supplemented with Plant Polyphenol Extracts

**DOI:** 10.3390/foods12091819

**Published:** 2023-04-27

**Authors:** Desalew Tadesse, Negussie Retta, Mekonnen Girma, Nicholas Ndiwa, Tadelle Dessie, Olivier Hanotte, Paulos Getachew, Dirk Dannenberger, Steffen Maak

**Affiliations:** 1Department of Animal Production and Welfare, Mekelle University, Mekelle 231, Ethiopia; tdesalew@gmail.com; 2Center for Food Science and Nutrition, Addis Ababa University, Addis Ababa 1176, Ethiopia; negussie.retta@gmail.com; 3LiveGene, International Livestock Research Institute (ILRI), Addis Ababa 5689, Ethiopia; m.girma@cgiar.org (M.G.); t.dessie@cgiar.org (T.D.); 4Research Methods Group, International Livestock Research Institute (ILRI), Nairobi 30709, Kenya; n.ndiwa@cgiar.org; 5Center for Tropical Livestock Genetics and Health (CTLGH), The Roslin Institute, Edinburgh EH25 9RG, UK; 6School of Life Sciences, University of Nottingham, Nottingham NG72UH, UK; 7Research Institute for Farm Animal Biology (FBN), 18196 Dummerstorf, Germany; dannenberger@fbn-dummerstorf.de (D.D.); maak@fbn-dummerstorf.de (S.M.)

**Keywords:** eggs, plant polyphenol extracts, flaxseed, slow-growing Sasso laying hens, n-3 PUFAs, lipid oxidative stability

## Abstract

Previous attempts to increase the level of flaxseed in hens’ diet for the production of n-3 polyunsaturated fatty acids (n-3 PUFAs)-enriched eggs have been commonly associated with undesirable effects on production efficiency, lipid health indices, and oxidative stability of eggs, requiring adequate research attention. This study investigated the effects of feeding a moderate level of flaxseed (FS) and plant polyphenol extracts (PPEs) on fatty acid content, oxidative stability, and lipid health indices in eggs of slow-growing Sasso T451A laying hens. One hundred and five hens were assigned to five groups (seven replicates of three) and fed on FS (75 g flaxseed and no antioxidants), VE8 (75 g flaxseed and 800 mg vitamin E), TS8 (75 g flaxseed and 800 mg *Thymus schimperi*), DA8 (75 g flaxseed and 800 mg *Dodonaea angustifolia*), and CD8 (75 g flaxseed and 800 mg *Curcuma domestica*) extract per kg diets. The egg yolk content of eicosapentaenoic acid (EPA, C20:5 n-3) in the DA8, TS8, and CD8 diets and docosahexaenoic acid (DHA, C22:6 n-3) in TS8 and CD8 diets significantly (*p* < 0.05) increased compared with the FS diet. The FS diet significantly increased the malondialdehyde (MDA) content in egg yolks, whereas the TS8 diet decreased it by 67% (*p* < 0.05). Little difference was observed in yolk fatty acid content between cooked and raw eggs. Production of n-3 PUFA-enriched eggs with favorable lipid health indices was possible through inclusion of PPEs extracted from local plant species grown in Ethiopia and a moderate dose of flaxseed in the diet of laying hens.

## 1. Introduction

Given the worldwide increase in cardiovascular diseases (CVD) and other chronic diseases, the impact of fat intake on human health has gained attention with the recommendation to increase the intake of long-chain n-3 polyunsaturated fatty acids (LC n-3 PUFAs) [1,2]. An important nutritional and health aspect linked to egg consumption is the high n-6 PUFA, mainly linoleic acid (LA: C18:2n-6) and limited n-3 PUFA content in eggs from laying hens fed conventional diet [3]. The high content of n-6 PUFAs in egg yolks is due to the greater inclusion of n-6 type vegetable fat in the hen’s diet. The LC n-3 PUFAs, such as eicosapentaenoic acid and docosahexaenoic acid (EPA, C20:5n-3, DHA, C22: 6n-3) are in short supply and deficient in the diet of most human populations [4]. The recommended daily intake of 250 mg of LC n-3 PUFAs, including at least 80 mg of EPA and DHA, remains as a challenge for most consumers [5,6,7].

To meet the recommended daily intake, enrichment of foods with LC n-3 PUFAs represents a solution [8]. Consumed by most people, the enrichment of chicken eggs with beneficial LC n-3 PUFAs through modification of dietary fat sources in hens’ diets is a promising method to deliver n-3 PUFAs, such as EPA and DHA to humans [9,10]. This approach has gained attention in the poultry and medical research communities [11].

Flaxseed (*Linum usitatissimum*) has been tested as a source of alpha-linolenic acid (ALA) precursor in the diet of laying hens for its conversion into long-chain unsaturated fatty acids such as EPA and DHA [12,13,14]. However, there are disadvantages in using flaxseed as a source of ALA to enrich eggs with n-3 fatty acids [15]. First, there is a limited conversion of ALA from flaxseed into EPA and DHA, and deposition in eggs remains scarce compared with the quantity of added flaxseed in the hens’ diets [9,16]. Second, the addition of a high level of flaxseed in laying hens’ diets was found to reduce oxidative stability in the diet and to increase the level of lipid peroxidation in eggs [17,18]. Increased oxidation in n-3 PUFA-enriched eggs reduces the nutritional value, develops an undesirable flavour, and produces toxic compounds [11,19].

Thus, the inclusion of a moderate dose of flaxseed (75 g/kg diet) alongside plant polyphenol extracts (PPE) in slow-growing Sasso T451A laying hens might represent a useful means to improve oxidative stability and to enhance n-3 PUFA content in eggs.

The use of natural antioxidants has been reported to increase the conversion capacity of added ALA into EPA and DHA [20,21], to enhance the antioxidant status in chickens [22], and to satisfy consumer concerns about the safety and toxicity of synthetic antioxidants [23]. Therefore, there is a growing interest in the application of polyphenol extracts from plants as natural antioxidants, which are better than synthetic antioxidant compounds. Natural polyphenols from different plants have been considered as an efficient way to prevent lipid peroxidation in eggs [15,24]. Polyphenol antioxidant properties contribute to a reduction in oxidative stress and protect and maintain the oxidation–reduction balance [25,26].

However, the inclusion of whole leaves/other plant parts in most previous studies has been associated with an increase in dietary fiber that is above the chickens’ requirement. This resulted in depressed performances of chickens [27,28] as they lack specific indigenous enzymes for fiber digestion [29], and one-fourth of their diet may be excreted in the feces undigested [30]. In addition, the inclusion of flaxseed at higher doses (>10%) in chicken diets was found to decrease the digestion and absorption of fats [31]. Hence, the inclusion of plant polyphenol extracts might improve the performances of hens and the n-3 PUFA content in eggs.

Studies have shown in a rat model that the inclusion of plant polyphenol extracts, such as curcumin, in combination with ALA-rich flaxseed oil, increased the DHA content in the brain [32], serum, and tissue [33]. A recent study reported an increase in desaturase enzyme activity in chickens fed with a combination of flaxseed and turmeric rhizome powder [34]. Moreover, improved oxidative stability in eggs has been reported in hens fed diets supplemented with bilberry and walnut leaves [24] and grape pomace or grape extracts [35].

Ethiopia has a great diversity of medicinal plants and spices which are used widely [36]. Among which, medicinal plants with higher phenolic contents, such as the sand olive (*Dodonaea angustifolia*), have been used as anti-plasmodial agents [37]. The sand olive possesses strong in vitro antioxidant activity to counteract oxidative stress [38]. Furthermore, thyme (*Thymus schimperi Ronniger*) has a high phenolic content, and therefore a strong antioxidant capacity and has been used to treat diabetic patients [39,40,41]. The other spice common in Ethiopian traditional kitchen is turmeric (*Curcuma domestica*) which has a high phenolic content and is an efficient antioxidant [42].

Dietary flaxseed to improve the n-3 PUFA content in eggs has been widely reported in layer-type chickens [20,43]. However, there is a dearth of studies dealing with slow-growing laying hens [44,45,46]. In this study, plant polyphenolic extracts of medicinal plants such as *D. angustifolia*, *T. schimperi*, and *C. domestica*, commonly found in Ethiopia, were hypothesized to represent an efficient source of antioxidant molecules in chicken diets to enhance n-3 PUFA content and prevent lipid peroxidation in egg yolks. Here, we assessed the effects of feeding flaxseed as a lipid source and plant polyphenol extracts as an antioxidant on n-3 PUFA content, oxidative stability, and the nutritional value of eggs. Previously, we reported a significant increase in health beneficial fatty acid contents in the raw breast muscle of slow-growing Sasso T451A chickens fed flaxseed combined with polyphenol extracts of *C. domestica* and *T. schimperi* compared to those fed flaxseed alone. Feeding chickens with *C. domestica* polyphenol extract and flaxseed maintained a high content of heat- and oxidative-susceptible fatty acids (DHA, EPA) in cooked breast compared to the flaxseed-only supplemented diet [46]. Thus, this study aimed to investigate the effects of dietary flaxseed as an ALA fat precursor source and polyphenolic extracts of *D. angustifolia*, *T. schimperi*, *C. domestica*, and alpha-tocopherol as antioxidants on n-3 PUFA content, oxidative stability, and lipid health indices in eggs of slow-growing Sasso laying hens.

## 2. Materials and Methods

### 2.1. Experimental Design, Diets, and Birds

The protocols for the study were approved by the Institutional Animal Care and Use Committee (IACUC2019-17.2) of the International Livestock Research Institute (ILRI). During the trial, chickens were managed according to the Sasso dual-purpose chicken management guide obtained from Hendrix Genetics (FGS, 2018, pp. 7–11) [47]. A total of 105 Sasso T451A hens were randomly assigned at 35 weeks into 5 different diets with 3 hens per pen (21 hens per treatment, 7 replicates of 3 hens) using a complete randomized block design. Study details, including diet composition and analysis, are provided in a previous publication [46]. To meet the nutritional needs of Sasso T451A chickens, a soybean corn-based diet was formulated using a rotating drum mixer. Diets were mixed in a rotating mixer for 30 min with plant extracts and vitamin E (α-tocopherol acetate) incorporated into a small quantity of wheat bran before being mixed with the main ingredients to ensure uniform dispersion throughout the feed. The experiment was conducted at the ILRI Poultry Research Facility in Addis Ababa, Ethiopia. Chickens were fed the following 5 diets for 8 weeks: FS (75 g flaxseed + no added antioxidant sources), VE8 (75 g of flaxseed + 800 mg α-tocopherol/kg), DA8 (75 g of flaxseed + 800 mg *D. angustifolia* extract/kg), TS8 (75 g of flaxseed + 800 mg of *T. schimperi* extract/kg), and CD8 (75 g of flaxseed + 800 mg of *C. domestica* extract/kg). Hens were fed 165 g feed/day. They were given free access to water, provided with 15 h of light during the egg-laying period, and were vaccinated against common diseases. The methods for the chemical composition and fatty acid analysis of the diets are outlined in our recent publication [46].

### 2.2. Chemicals and Materials

Chemicals: Vitamin E (α-tocopherol acetate, Shaanxi Bolin Bio-tec, Xian, China), stearidonic acid (C18:4n-3, Larodan, Limhamn, Sweden), the QuantiChrom TBARS assay kit (DTBA-1000, BioAssay Systems, Hayward, CA, USA), and conjugated linoleic acid (C18:2cis-9, trans-11, Matreya, State College, PA, USA) were purchased from different suppliers. ABTS (2, 2′-azinobis-3-ethylbenzothiazoline-6-sulfonic acid), DPPH, (2, 2-diphenyl-1-picrylhydrazyl), gallic acid, Trolox (6-hydroxy-2, 5, 7, 8-tetramethyl-chroman-2-carboxylic acid), fatty acid methyl esters (FAME), and adrenic acid (C22: 4n-6) were purchased from Sigma-Aldrich, Germany.

Materials: Bulk beads (zirconium oxide Precellys beads) and the homogenizer (Precellys Evolution) were purchased from Bertin Instruments Technologies, Montigny-le-Bretonneux, France. Sarstedt tubes (Adelab Scientific, Thebarton, Australia), Digital kitchen thermometer (DOQAUS, Futian District Shenzhen, China), a centrifuge (ScanSpeed 40, LaboGene, Allerd, Denmark), a lab mill (Model A11 Basic, IKA GmbH, Staufen, Germany), a microwave plasma-atomic emission spectrometer (Agilent Technologies, 4200 MP-AES, Santa Clara, CA, USA), Pyrex tubes (Pyrex, Hayes, UK), CP-Sil 88 CB columns (100 m × 0.25 mm, Agilent, Santa Clara, CA, USA), a PerkinElmer gas chromatograph CLARUS 680 (PerkinElmer Instruments, Waltham, MA, USA), a 96-well plate for the plate reader (SynergyTM MX, BioTek, Bad Friedrichshall, Germany), food saver vacuum bags (Food Saver, PN:192465, Republic of Korea) and Water bath (Clifton, Nickel Electro Ltd.,Oldmixon Crescent, Weston-super-Mare, UK, GB) were purchased from various suppliers.

### 2.3. Antioxidant Capacity of Plant Polyphenol Extracts (PPEs) and Diets

The methods used to prepare plant and diet extracts and determine the phenolic content and antioxidant power are described in detail in our recent work [46]. The total phenolic content of *T. schimperi*, *D. angustifolia*, and *C. domestica* was 72.34, 260.82, and 277.64 mg GAE/g, respectively. The PPE from the 3 plants had IC_50_ values of 33.97, 25.59, and 96.98 µg/mL to scavenge DPPH free radicals. The respective ABTS+ free radical-scavenging IC_50_ values were 6.75, 8.81, and 13.45 µg/mL. Flavones and luteolin are the major polyphenol components in *T. schimperi*, comprising up to 46.05 g/kg of the fresh weight [48]. Curcumin is the major polyphenol (>71%) found in turmeric (*C. domestica*) rhizomes [49]. The *D. angustifolia* is considered as a rich source of phenols (27.5 mg/g), steroids (17.7 mg/kg), and flavonoids (20.5 mg/g) in the dry extract weight [49].

### 2.4. Egg Sample Collection and Processing

Eggs of similar weight were collected in the eighth week of the experimental period (14 per treatment) and stored at +4 °C until further processing. To collect raw yolk samples, eggs were carefully broken, and the yolks were separated from the albumen by gently rolling on filter paper. Then, 20 g of raw yolk sample was taken and kept in marked Sarstedt tubes at −18 °C until analysis. Cooked egg samples were prepared according to Nimalarante and co-workers [50]. Accordingly, eggs were taken from the refrigerator and maintained at room temperature for 12 h before cooking. After adding water up to 3–5 cm above the eggs in a single-layer pan, eggs were cooked in heated water at 76 °C for 10 min. The applied temperature was monitored using the DOQAUS digital kitchen thermometer. After the cooked eggs were cooled under running tap water for 5 min, the yolk was carefully separated from the white. Then, from the cooked eggs, 20 g of yolk sample was taken and kept in marked Sarstedt tubes and stored at −18 °C until analysis.

### 2.5. Fatty Acid Analysis

The egg yolk sample preparation for fatty acid analysis was carried out as recently detailed for chicken breast muscle with only a few modifications [46].

#### 2.5.1. Lipid Extraction and Transesterification of Raw and Cooked Egg Yolks

The raw and cooked frozen egg yolk samples were homogenized manually. For lipid extraction, approximately 2 g of the egg yolk was weighed. Each Precellys tube contained 20 pieces of 2.8 mm bulk beads and 2 pieces of 5 mm bulk beads. After the addition of 3 mL of methanol and non-adeconoic acid (19:0) as the internal standard, the extracts (in duplicate) were homogenized 3 times at 25 s intervals at 4 °C and 6500 rpm using a homogenizer [51]. After they were vortexed, the homogenates were transferred to Pyrex tubes containing 8 mL chloroform. The Precellys tubes were then rinsed twice with 1 mL of methanol. To prevent oxidation of PUFA, all solvents used for egg yolk lipid extraction contained 0.005% (*w*/*v*) of t-butylhydroxytoluene (BHT).

After filtration, the lipid extracts of yolk samples were stored at 5 °C for 18 h in the dark and subsequently washed with a 0.02% CaCl_2_ solution. The organic phase was separated and dried using a mixture of Na_2_SO_4_ and K_2_CO_3_ (10:1, *w/w*), and the solvent was subsequently removed using a vacuum centrifuge at 438× *g* at 30 °C for 30 min. The lipid extracts (25 mg aliquot) were redissolved in 300 μL of toluene for the preparation of methyl ester [52]. Then, to transmethylate lipid extracts, 2 mL of 0.5 M sodium methoxide in methanol was added and agitated for 10 min at 60 °C in a water bath. Subsequently, 1 mL of 14% boron trifluoride in methanol was added to the mixture, which was then shaken for another 10 min at 60 °C. The fatty acid methyl esters (FAMEs) were extracted twice with 2 mL of n-hexane and stored at −18 °C until high-resolution gas chromatography (HR-GC) analysis.

#### 2.5.2. Gas Chromatography Analysis

Capillary high-resolution gas chromatography (HR-GC) with a CP-Sil 88 CB column was used to analyze the fatty acids in all sample extracts. The GC column was installed on a PerkinElmer gas chromatograph CLARUS 680 equipped with a flame ionization detector and split injection, as previously described [53]. In brief, hydrogen was employed as the carrier gas at a flow rate of 1 mL × min^−1^ while the split ratio was 1:20, with the injector and detector adjusted at 260 °C and 280 °C, respectively. The GC oven temperature programme was 150 °C for 5 min; 2 °C/min heating until 200 °C and held for 10 min; and 1 °C/min heating until 225 °C and held for 20 min. To calibrate the reference standard mixture “Sigma FAME”, the methyl esters of C18:1cis-11, C22:5n-3, and C18:2cis-9, and trans-11, C22:4n-6, and C18:4n-3 were employed. After GC analysis of 5 samples, the 5-point calibration of single fatty acids assessed were ranged between 16 and 415 µg/mL. Fatty acid concentrations were displayed in milligrams per gram of egg yolk.

### 2.6. Analysis of Oxidative Stability in Egg Yolks

The analysis of oxidative stability in egg yolk samples was performed as recently described for chicken muscle [46]. In short, frozen egg yolk samples were homogenized and approximately 400 mg of yolk sample was weighed in a tube for oxidative stability analysis. Each 7 mL Precellys tube contained 5 pieces of 2.8 mm bulk beads and 1 piece of 5 mm bulk beads. Sample preparation and oxidative stability measurements in egg yolk samples were carried out in accordance with the QuantiChrom TBARS assay kit guidelines. After adding 2 mL of ice-cold phosphate-buffered saline (PBS) solution (200 mL and 133 µL BHT), the extracts (in triplicate) were homogenized twice at 10-s intervals at 4 °C and 6500 rpm using a homogenizer.

Then, 200 µL of ice-cold 10% trichloroacetic acid (TCA) solution was added and incubated for 5 min on ice. Following this, the sample extracts were centrifuged for 5 min at 14,000× *g*, and 200 µL of thiobarbituric acid (TBA) was added to 200 µL sample extract, vortexed, and incubated for 60 min at 100 °C. After cooling to room temperature, 100 µL of the sample extracts were transferred into the 96-well plate of the plate reader. The color intensity (OD) was measured at 535 nm. The malondialdehyde (MDA) standard was developed from a concentration range of 0.0 to 1.5 µM MDA, and the color intensity was assessed using the same approach as for sample solutions. Using MDA standard calibration, the concentrations of reactive thiobarbituric acid substances (TBARS) were calculated. Finally, the TBARS concentration was given in µg MDA/g of egg yolk.

### 2.7. Calculating Lipid Health Indices

The potential health contribution of dietary lipids may be shown by estimating the relationship between individual fatty acids and their groups within the major class of unsaturated fatty acids [20]. In light of this, the hypocholesterolemic/hypercholesterolemic (h/H) ratio, atherogenic indices (AI), thrombogenic indices (TI), saturation indices (s/p), and the n-6/n-3 PUFA ratio for egg yolk lipids were evaluated. The AI, TI, and h/H ratios were calculated according to Equations (1)–(3) [53]. Equations (4)–(7) were used to calculate the egg yolk PUFA ratio, s/p ratio, the total sum of fatty hypercholesterolemic fatty acids (HFA), desirable fatty acids (DFA), and nutritional value indices (NVI) [20].
(1)$$AI=\frac{\left(4\times C14:0+C12:0+C16:0\right)}{\left[\sum MUFA+\sum n6PUFA+\sum n3PUFA]\right]}$$
(2)$$TI=\frac{\left(C14:0+C16:0+C18:0\right)}{\left[\left(0.5*MUFA\right)+(0.5*n-6PUFA+\left(3*n-3PUFA\right)+\left(\frac{n-3PUFA}{n-6PUFA}\right)\right]}$$
(3)$$\begin{array}{l} h/H\\ =\frac{\left(C18:1n9+C18:2n6+C18:3n3+C20:4n6+C20:5n3+C22:5n3+C22:6n3\right)}{\left(C14:0+C16:0\right)} \end{array}$$
(4)$$s/p=\frac{\left(C14:0+C16:0+C18:0\right)}{\left(MUFA+PUFA\right)}$$
(5)HFA=C12:0+C14:0+C16:0
(6)DFA=C18:0+UFA
(7)$$NVI=\frac{C18:0+C18:1}{C16:0}$$
where: PUFA-polyunsaturated fatty acids; MUFA-monounsaturated fatty acids; UFA-unsaturated fatty acids.

### 2.8. Statistical Analysis

The data analysis was undertaken as per the statistical method that was used for chicken breast muscle data with only a few modifications [46]. Data were analyzed using the R platform [54] and the RStudio environment for statistical computation [55]. The “lme4” package was applied to perform a mixed linear model [56]. A linear mixed model was fitted by using a pen as a random effect and room and diet as fixed effects to predict the egg yolk fatty acid concentrations as a response variable. Using the R plot function, the residual distributions were examined visually to evaluate the underlying assumptions of the analysis (i.e., linearity, independence, homoscedasticity, normality of residuals). The estimated marginal means were produced using the “emmeans” package [57] for each diet with 95% confidence intervals. Using Tukey’s adjusted *p*-values, multiple paired comparisons between treatments were carried out to look for overall significant effects.

## 3. Results and Discussion

### 3.1. Fatty Acid Content in Raw Egg Yolks

Table 1 reports the fatty acid content of the raw egg yolk, and the values for all investigated fatty acids are presented in Appendix A. In the current investigation, egg yolk fat percentage, saturated fatty acids (SFAs) including myristic (C14: 0), palmitic (C16: 0), and stearic (C18: 0), and monounsaturated fatty acid (MUFA) contents did not change (*p* > 0.05) between the treatment groups. In line with the present study, hens fed various types of flaxseed as a dietary source of fat had no influence on egg yolk fat percentage as reported by Sosin et al. [58]. The present findings of no effect on egg yolk, SFA, and MUFA contents agreed with similar observations in hens fed with 5% oil and marigold extract [59] and flaxseed meal with either 2% dried kapia pepper, sea buckthorn pomace, or carrot [11]. In contrast, a decrease in the content of SFAs in egg yolk was reported in hens fed with flaxseed and *Rhus coriaria* seed or *Zingiber officinale* root powder as antioxidant sources [60]. Moreover, our results of vitamin E on egg yolk SFAs and MUFAs in this study agreed with the previous report by [61].

In the present study, the total n-6 PUFA, linoleic acid (LA, C18:2n-6), and arachidonic acid (AA, C20: 4n-6) contents in egg yolks were not affected by feeding hens with flaxseed and/or with plant extracts or vitamin E (*p* > 0.05). Similarly, no effect on egg yolk PUFA, AA and LA contents was reported in hens fed flaxseed either with *Rosa canina* meal [62], dried kapia pepper, sea buckthorn, or carrots [11]. In contrast, inhibitory effects on the synthesis of n-6 PUFAs were reported upon the inclusion of dietary ALA at high doses [63,64].

Compared to the FS diet, the yolk ALA content increased by 28.9% in the TS8 diet, 22% in the CD8 diet, and 15.6% in the DA8 diet (*p* > 0.05). One advantage of flaxseed inclusion over fish oil/meal in hens diet is considered egg enrichment with ALA, eicosapentaenoic acid (EPA, C20:5n-3), and docosahexaenoic acid (DHA, C22:6n-3) [65]. Consistent with the present results, an increase in egg yolk ALA content was reported in hens fed flaxseed either with dried kapia pepper, sea buckthorn pomace, or carrot [11] and thyme extract [66] as antioxidant sources. In addition, no effect was observed from vitamin E on egg yolk ALA content in this study, which agrees with the report of Botsoglou and co-workers [67] in hens fed 4% linseed oil with levels of olive leaves meal and vitamin E as antioxidant sources.

In the current findings, hens fed polyphenol extracts were found to be more efficient in increasing the most potent health-promoting fatty acids, such as EPA and DHA, in egg yolks than diets with sole supplementation of flaxseed. Consequently, compared to the FS diet, hens fed the DA8, TS8, and CD8 diets showed significantly increased levels of EPA (*p* < 0.05), and the TS8 and CD8 diets increased (*p* < 0.05) DHA in egg yolks. Other studies have found a considerable increase in the egg yolk DHA in hens given 10% flaxseed either with pine wood polyphenols or lutein antioxidant sources [68] and 9% flaxseed meal along with sea buckthorn meal [20].

In this study, compared to the FS diet, feeding hens with plant extracts in the TS8 and CD8 diets improved the total n-3 PUFA content in egg yolks (*p* < 0.05). Increasing the sum of n-3 PUFAs (ALA + EPA + DPA + DHA) in egg yolks is one of the most desirable functions of enriched eggs, making them more favorable than regular eggs for human nutrition [69,70]. In agreement with the present study, Vlaicu et al. found an increase in the egg yolk n-3 PUFA content in hens fed flaxseed along with *Rosa canina* meal as a natural antioxidant source compared to sole supplementation of flaxseed [62]. Moreover, a similar increase in egg yolk total n-3 PUFA content was observed in hens fed 10% flaxseed along with polyphenols and lutein natural antioxidant sources [68]. In addition, feeding hens flaxseed along with holy basil (*Oscimum sanctum*) was found to increase the n-3 PUFA content in egg yolks [71].

Furthermore, feeding hens with plant extracts markedly increased the egg yolk content of EPA by 25% in the DA8 diet and 31% in both the TS8 and CD8 diets compared to vitamin E in the VE8 diet (*p* < 0.05). The egg yolk DHA content increased (*p* < 0.05) in the TS8 diet and slightly increased (*p* > 0.05) in the DA8 and CD8 diets compared to hens fed the VE8 diet. Consistent with this study, a previous study found that canthaxanthin supplementation as a natural antioxidant was more effective than vitamin E in increasing DHA content in egg yolks [72]. Moreover, no difference in egg yolk n-3 PUFA content between hens fed plant polyphenol extracts and vitamin E was observed in our study, which is in line with the report of Vakili and co-workers [67] in hens fed 200 mg of vitamin E and linseed oil. In contrast, few studies have reported a decreasing effect on egg yolk n-3 PUFA content in hens fed levels of vitamin E along with 3% fish oil [73] and 5% linseed oil [74].

The potential contribution of consuming an egg in meeting the recommended daily amount of n-3 PUFAs was estimated. Accordingly, the contents of ALA, EPA, DPA, and DHA in an egg with a 19 g yolk weight provides 314 mg of n-3 PUFAs in the CD8 diet and 326 mg in the TS8 diet. To be labeled as omega-3, eggs need to contain at least 300 mg/60 g of egg in a country such as Canada [75]. More importantly, the required 300 mg of n-3 PUFA was met in eggs from hens fed 7.5 g flaxseed alongside 800 mg polyphenol extracts in the CD8 and TS8 diets. Moreover, estimating the most potent LC n-3 PUFAs shows that consuming an egg with a 19 g yolk weight provides 3.99 mg of EPA and 140 mg of DHA in the TS8 diet and 3.99 mg of EPA and 131 mg DHA in CD8-fed hens, contributing to 28 to 48% and 27 to 45% of the daily recommended human intake (300 to 500 mg of EPA and DHA), respectively.

The increase in EPA and DHA in egg yolks observed in our study could be explained by the prevention of degradation of n-3 PUFAs by plant polyphenol extracts. The polyphenol extracts also improved the conversion efficiency of ALA to EPA and DHA by activating the lipid-metabolizing liver enzymes, as Huang et al. reported [76]. Furthermore, plant extracts might enhance the antioxidant capacity of laying hens in vivo, as similar effects were observed in hens fed with polyphenols from marigold extract with an added oil source [59]. However, further studies are required to confirm the extent to which lipid-metabolizing enzymes are enhanced (enzyme activity/expression) in the liver.

### 3.2. Fatty Acid Content in Cooked Egg Yolks

The cooked egg yolk fatty acid contents are shown in Table 2, and values for all investigated fatty acids are provided in Appendix A. This study evaluated the extent of fatty acid loss and explored the possible protective effects of polyphenol extracts on heat-sensitive and oxidative-susceptible eggs. According to our findings, the fat composition, SFA, and MUFA content in eggs cooked at 76 °C for 10 min did not vary between the treatment groups (*p* > 0.05). Similarly, the total n-6 in the cooked egg yolk did not change across treatments, although cooking marginally decreased (*p* > 0.05) the AA content (C20:4n-6) and increased (*p* > 0.05) the LA content (C18:2n-6) in the egg yolks of hens fed plant extracts compared to the FS diet. In support of our findings, some earlier research did not find an influence on SFA and MUFA in enriched eggs [77] either cooked at 100 °C for 30 min [78] or 100 °C for 4 to 15 min [79]. Another study also found an increase in SFA, but no effect on MUFA in eggs cooked at 100 °C for 12 min [80]. Table 2 shows the effects of feeding plant polyphenol extracts and flaxseed on fat content and fatty acid concentrations in the yolk of cooked eggs of slow-growing Sasso laying hens.

In this study, feeding with polyphenol extracts showed a protective effect on heat-sensitive and oxidative-susceptible fatty acids in egg yolks. Compared to the FS diet, hens fed on plant extracts (DA8, TS8, and CD8 diets) maintained a higher (*p* < 0.05) ALA level in cooked egg yolks. Furthermore, hens fed the TS8 diet retained higher DPA, EPA, and DHA contents in cooked egg yolks than those fed FS and VE8 diets (*p* < 0.05). Moreover, hens fed the TS8 diet retained 16% and 10% more PUFA content in cooked egg yolks compared to the FS and VE8 diets, respectively (*p* < 0.05). Furthermore, feeding hens with plant extracts retained (*p* < 0.05) more n-3 PUFAs (+18% in CD8, +19% in DA8 and +31% in TS8) in cooked egg yolks compared to the FS diet. Among plant extracts, the TS8 diet (but not the DA8 and CD8 diets) significantly (*p* < 0.05) increased the n-3 PUFA content in cooked egg yolk compared to the VE8 diet.

Previously, studies evaluated differences in fatty acid content between raw and cooked eggs [70,79]. Studies evaluating the fatty acid content in cooked eggs of hens fed diets supplemented with different natural antioxidants are limited. Here, we observed an increase in the levels of EPA and DHA in cooked eggs from hens fed with plant extracts, possibly following the transfer of bioactive polyphenol antioxidants from the chicken to the eggs. These bioactive polyphenols are known to increase the overall antioxidant status of laying hens and protect heat-sensitive n-3 PUFAs in eggs during cooking [81]. Consistent with the present results, slight changes in the PUFA content and an increase in n-3 PUFA were found when cooking at 75 °C for 20 min, but these values decreased when the temperature was raised to 85 °C [7].

Overall, the current findings showed little difference in the fatty acid content between raw and cooked yolks. Similarly, previous studies found no difference between yolk fatty acid content in raw and cooked enriched eggs [70,82]. Enriched eggs were also found to be relatively stable during cooking in the presence of antioxidants [77]. Furthermore, these small differences observed between raw and cooked eggs could be associated with either the protective effect of plant polyphenol extracts or the presence of the vitelline membrane, albumen, and shell surrounding the yolk, which prevented water loss, diffusion of gases, and lipid oxidation during cooking of shelled eggs. These factors are likely to preserve the fatty acids in the egg yolk from possible damage during cooking, as has been well stated in previous studies [7,77].

### 3.3. Lipid Health Indices in Cooked and Raw Egg Yolks

Results for oxidative stability and lipid health indicators in raw and cooked egg yolks are shown in Table 3. In the current study, adding plant extracts to the diet decreased the ratio of n-6/n-3 PUFA from 3.07 in the FS diet to 2.61 (−17%) in the DA8 diet, 2.54 (−20%) in the TS8 diet, and 2.52 (−21%) in the CD8 diet (*p* > 0.05). These values satisfied the recommended n-6/ n-3 PUFA ratio of 4:1 to 1:1 for desirable human health benefits [83]. The increase in egg yolk n-3 PUFAs with a concurrent decrease in n-6 PUFA content in hens fed plant extract-supplemented diets decreased the egg yolk n-6/n-3 PUFA ratio. A slight decrease in the PUFA ratio in hens fed with 4% linseed oil and with 5 g or 10 g of olive leaves as a source of polyphenol antioxidants was previously reported [67]. In cooked egg yolks, the n-6/n-3 PUFA ratios were decreased by −10% on average in diets supplemented with plant extracts compared to the flaxseed diet. Overall, the n-6/n-3 PUFA ratios were lower in raw yolks than in cooked yolks, which could be associated with a slight increase in n-6 PUFA in cooked eggs compared to raw egg yolks. Table 3 shows the lipid health indices and malondialdehyde (MDA) in egg yolks of Sasso laying hens.

In this study, while there was no difference (*p* > 0.05) in the egg atherogenic indices (AI) and thrombogenic indices (TI) between treatment groups, feeding hens with plant extracts reduced the TI indices by 4% in the DA8 diet, 6.6% in the CD8 diet, and 9.3% in the TS8 diet compared to the FS diet. Healthy animal products such as eggs with desirable health benefits are defined by lower values of AI and TI and higher hypocholesterolemic/hypercholesterolemic (h/H) indices [53]. An increase in TI and AI could indicate a decline in the nutritional quality of egg yolks [84], with the optimal value suggested to be less than 1.00 [85]. In summary, the indices observed here of less than 1.00 AI (0.50 to 0.53) and TI (0.68 to 0.75) satisfy the recommended range for perceived cardiovascular health benefits [85]. Consistently similar values in TI, AI, and s/p ratios were reported by [20] in hens fed with 9% flaxseed and 3% sea buckthorn. Interestingly, the higher h/H ratios observed in this study may indicate better nutritional quality than those reported in eggs from similar dietary protocols [20].

In cooked eggs, the AI indices and h/H ratios did not differ between the dietary treatment groups. However, the TI indices in cooked eggs were reduced in DA8 and TS8 diets compared to the FS diet (*p* < 0.05). Additionally, there was no difference (*p* > 0.05) found in cooked egg TI indices between hens fed on plant extracts and vitamin E. A similar h/H ratio but with relatively lower AI and higher TI indices was reported by [80] in cooked eggs boiled at 100 °C for 12 min. Furthermore, the nutritional value indices (NVI), hypercholesterolemic fatty acids (HFA), and saturation indices (s/p) did not vary (*p* > 0.05) between the treatment groups. Except for slight differences in NVI, the n-6/n-3 ratio, and HFA, no differences were observed in lipid health indices between raw and cooked yolks. The lack of differences in lipid heath indices between raw and cooked yolks might be associated with the lipid peroxidation prevention effect of plant polyphenol extracts during cooking.

### 3.4. Oxidative Stability in Egg Yolks

The results of egg yolk lipid peroxidation are presented in Table 3. In this study, the oxidative stability of enriched eggs from hens fed on plant polyphenol extracts or vitamin E along with flaxseed was investigated. Among the plant extracts, the TS8 diet was the most efficient (*p* < 0.05) in reducing the egg yolk MDA content by 67% compared to the FS diet. The decrease in egg yolk MDA content was followed by 12% in the CD8 diet and 29% in the DA8 diet (*p* > 0.05) compared to the FS diet. Here, the observed higher MDA level in hens fed with the FS diet reflects the intensity of lipid peroxidation in diets not supplied with dietary antioxidants. Flaxseed inclusion as a fat source in a hen’s diet has been found to enhance the LC n-3 PUFA content but causes a significant decrease in the oxidative stability of eggs [86,87].

The observed decrease in egg yolk MDA content in the present study could be due to the plant polyphenol extracts that prevented oxidative stress associated with the intake of diets enriched with PUFA or the in vivo decreased production of oxidative products, as stated in previous studies [46,88]. Consistently, recent studies have reported a decreasing effect on egg yolk lipid peroxidation in hens fed flaxseed along with natural antioxidants, such as pine wood polyphenols or lutein [68], dried kapia pepper, sea buckthorn pomace, and dried carrot [11]. In a similar recent study, a significant decrease in egg yolk MDA was reported in laying hens fed 60 g /kg grape pomace, but grape extract has lower antioxidant potential compared to grape pomace [35].

In this study, feeding hens with a high dose of vitamin E had no significant effect (*p* > 0.05) on egg yolk MDA content compared to the FS diet. Furthermore, in the TS8 diet, the egg yolk MDA content was reduced by 65% compared to the VE8 diet. Similarly, the addition of plant extracts reduced the MDA content of the egg yolk by 25% in the DA8 diet and 7% in the CD8 diet compared to the VE8 diet (*p* > 0.05). Overall, the incorporation of vitamin E in the feed was not as effective as the plant polyphenol extracts in reducing the egg yolk MDA content. Similar effects were observed in chicken breast muscle in our recent study [46]. Previous studies recommended the inclusion of higher doses of vitamin E, as lower doses in a diet enriched with n-3 PUFAs were found to be inefficient in maintaining the oxidative stability of eggs [89]. In contrast, vitamin E was equally as effective as natural antioxidants [68], found to be more effective than olive leaves meal [67], and more effective than rosemary extract [90] in reducing the MDA content in egg yolk.

## 4. Conclusions

Adding plant polyphenol extracts along with a moderate dose of flaxseed (7.5 mg/kg) in hens’ diets can be considered a useful approach to increase the egg yolk content of n-3 PUFAs, such as eicosapentaenoic acid (EPA) and docosahexaenoic acid (DHA), while maintaining favorable atherogenic and thrombogenic indices and a suitable n-6/n-3 PUFA ratio for consumer health. Plant polyphenol extracts were found to protect heat-sensitive and oxidative-prone fatty acids during cooking. The deteriorated oxidative stability of fatty acids because of flaxseed feeding was notably improved by the inclusion of 800 mg *Thymus schimperi* and 800 mg *Dodonaea angustifolia* polyphenol extracts. Inclusion of 800 mg α-tocopherol (vitamin E) in laying hens’ diets slightly decreased the MDA content in the egg yolk, but it was not efficient compared to feeding either 800 mg *T. schimperi*, *D. angustifolia*, or *Curcuma domestica* polyphenol extracts. The findings of this study offer new opportunities to enrich egg yolks with beneficial fatty acids using local sources and/or indigenous plants for the benefit of local consumers. Furthermore, the isolation of bioactive compounds that play an antioxidant role in the extracts should be investigated. The consumer acceptability of eggs from laying hens fed plant extracts should be evaluated.

## Figures and Tables

**Table 1 foods-12-01819-t001:** The effects of using plant polyphenol extracts and flaxseed on fat content and fatty acid concentrations in yolks of raw eggs from slow-growing Sasso laying hens.

Fatty Acids (mg/g)	^1^ Treatments										Random Effect	*p*-Value
	FS		VE8		DA8		TS8		CD8			
	Mean	95% [LCL, UCL]	Mean	95% [LCL, UCL]	Mean	95% [LCL, UCL]	Mean	95% [LCL, UCL]	Mean	95% [LCL, UCL]		
C14:0	0.80	[0.72, 0.86]	0.78	[0.70, 0.85]	0.82	[0.74, 0.89]	0.77	[0.69, 0.84]	0.80	[0.73, 0.87]	non 0	0.8556
C16:0	63.90	[61.1, 66.7]	65.20	[62.4, 68.0]	66.30	[63.6, 69.1]	64.00	[61.3, 66.8]	65.50	[62.7, 68.3]	non 0	0.6833
C18:0	22.70	[21.6, 23.9]	21.50	[20.3, 22.6]	22.20	[21.0, 23.3]	23.10	[21.9, 24.2]	22.70	[21.5, 23.9]	0	0.3320
C14:1cis-9	0.15	[0.12, 0.18]	0.14	[0.11, 0.17]	0.17	[0.14, 0.20]	0.14	[0.11, 0.17]	0.16	[0.13, 0.19]	non 0	0.4706
C16:1cis-9	6.97	[6.17, 7.78]	6.68	[5.87, 7.49]	7.57	[6.76, 8.38]	6.58	[5.77, 7.39]	7.31	[6.50, 8.12]	non 0	0.3638
C18:2*n*-6	33.70	[30.9, 36.5]	37.20	[34.4, 40.1]	35.60	[32.8, 38.4]	38.8	[36.0, 41.6]	36.20	[33.4, 39.1]	non 0	0.1383
C18:3*n*-3	6.60	[5.39, 7.80]	6.56	[5.36, 7.77]	7.63	[6.43, 8.83]	8.51	[7.31, 9.72]	8.08	[6.88, 9.28]	non 0	0.0981
C20:4*n*-6	3.79	[3.52, 4.07]	4.09	[3.81, 4.37]	3.65	[3.37, 3.93]	4.09	[3.81, 4.37]	4.00	[3.73, 4.28]	non 0	0.1045
C20:5*n*-3	0.16 ^a^	[0.14, 0.18]	0.16 ^a^	[0.13, 0.17]	0.20 ^b^	[0.17, 0.22]	0.21 ^b^	[0.18, 0.23]	0.21 ^b^	[0.19, 0.23]	0	0.0003
C22:5*n*-3	0.81 ^a^	[0.60, 1.02]	0.95 ^a^	[0.75, 1.16]	1.29 ^b^	[1.08, 1.50]	1.07 ^ab^	[0.86, 1.27]	1.08 ^ab^	[0.87, 1.29]	non 0	0.0338
C22:6*n*-3	5.96 ^a^	[5.51, 6.42]	6.31 ^ab^	[5.86, 6.77]	6.50 ^ab^	[6.04, 6.95]	7.39 ^c^	[6.93, 7.84]	6.93 ^bc^	[6.47, 7.38]	non 0	0.0011
^2^ ∑SFA	88.20	[84.9, 91.6]	88.40	[85.1, 91.7]	90.20	[86.9, 93.5]	88.80	[85.5, 92.1]	89.90	[86.5, 93.2]	non 0	0.8785
^3^ ∑MUFA	117	[113, 122]	116	[112, 120]	118	[114, 122]	117	[113, 121]	119	[115, 123]	0	0.9255
^4^ ∑ PUFA	53.10 ^a^	[49.2, 56.9]	57.50 ^ab^	[53.6, 61.3]	57.00 ^ab^	[53.1, 60.8]	62.10 ^b^	[58.3, 66.0]	58.50 ^b^	[54.7, 62.4]	non 0	0.0366
∑ n-3 PUFAs	13.70 ^a^	[12.1, 15.3]	14.20 ^a^	[12.6, 15.8]	15.80 ^ab^	[14.2, 17.4]	17.40 ^b^	[15.8, 19.0]	16.50 ^b^	[14.9, 18.1]	non 0	0.0122
∑ n-6 PUFAs	39.10	[36.3, 41.9]	43.10	[40.3, 45.9]	40.90	[38.0, 43.7]	44.50	[41.6, 47.3]	41.80	[38.9, 44.6]	non 0	0.0935
Fat (%)	25.90	[25.2, 26.5]	26.20	[25.5, 26.9]	26.50	[25.8, 27.2]	26.80	[26.1, 27.5]	26.70	[26.0, 27.4]	non 0	0.2900

Estimated marginal means (emmeans) with 95% upper confidence (UCL) and lower confidence limits (LCL) (n = 70); emmeans within a row with ^a, b, c^ superscript letters are significantly different at *p* < 0.05. ^1^ FS: 75 g of flaxseed + no antioxidant sources, VE8: 75 g of flaxseed + 800 mg α-tocopherol/kg, DA8: 75 g of flaxseed + 800 mg of *D. angustifolia* extract/kg, TS8: 75 g of flaxseed + 800 mg of *T. schimperi*/kg, CD8: 75 g of flaxseed + 800 mg of *C. domestica*/kg diet. ^2^ Sum of saturated fatty acids. ^3^ Sum of monounsaturated fatty acids. ^4^ Sum of polyunsaturated fatty acids.

**Table 2 foods-12-01819-t002:** The effects of feeding plant polyphenol extracts and flaxseed on fat content and fatty acid concentrations in the yolk of cooked eggs from slow-growing Sasso laying hens.

Fatty Acids (mg/g)	^1^ Treatments										Random Effect	*p*-Value
	FS		VE8		DA8		TS8		CD8			
	Mean	95% [LCL, UCL]	Mean	95% [LCL, UCL]	Mean	95% [LCL, UCL]	Mean	95% [LCL, UCL]	Mean	95% [LCL, UCL]		
C14:0	0.79	[0.71, 0.85]	0.81	[0.74, 0.87]	0.79	[0.72, 0.85]	0.83	[0.76, 0.89]	0.81	[0.74 0.87]	non 0	0.8464
C16:0	63.90	[61.1, 66.7]	65.60	[62.9, 68.4]	63.70	[61.0, 66.5]	66.40	[63.7, 69.2]	65.60	[62.8, 68.3]	non 0	0.5566
C18:0	21.70	[20.6, 22.8]	22.60	[21.5, 23.7]	22.10	[21.0, 23.2]	22.30	[21.2, 23.4]	23.60	[22.5, 24.7]	non 0	0.16807
C14:1cis-9	0.16	[0.13, 0.18]	0.16	[0.12, 0.18]	0.16	[0.12, 0.18]	0.17	[0.13, 0.19]	0.16	[0.12, 0.18]	non 0	0.981
C16:1cis-9	7.07	[6.27, 7.87]	6.99	[6 19, 7.80]	6.91	[6.11, 7.71]	7.43	[6.63, 8.24]	7.00	[6.19, 7.80]	non 0	0.8927
C18:2*n*-6	34.60	[31.2, 38.0]	35.50	[32.1, 38.9]	36.40	[33.0, 39.8]	38.90	[35.5, 42.3]	36.00	[32.6, 39.4]	non 0	0.4476
C18:3*n*-3	6.42 ^a^	[5.44,7.39]	7.22 ^ab^	[6.25, 8.19]	8.16 ^b^	[7.19, 9.12]	8.48 ^b^	[7.51, 9.45]	7.99 ^b^	[7.02, 8.96]	non 0	0.03423
C20:4*n*-6	4.01	[3.73, 4.29]	3.97	[3.69, 4.25]	3.65	[3.37, 3.93]	4.15	[3.87, 4.43]	3.82	[3.54, 4.10]	non 0	0.1302
C20:5*n*-3	0.17 ^a^	[0.13, 0.19]	0.17 ^a^	[0.14, 0.20]	0.19 ^ab^	[0.16, 0.22]	0.23 ^b^	[0.20, 0.26]	0.18 ^ab^	[0.15, 0.21]	non 0	0.0165
C22:5*n*-3	0.83 ^a^	[0.63, 1.02]	1.26 ^b^	[1.07, 1.45]	1.05 ^ab^	[0.86, 1.24]	1.44 ^c^	[1.25, 1.63]	1.05 ^ab^	[0.86, 1.24]	non 0	0.00095
C22:6*n*-3	6.06 ^a^	[5.70, 6.42]	6.74 ^b^	[6.38, 7.11]	6.56 ^ab^	[6.20, 6.93]	7.45 ^c^	[7.09,7.81]	6.67 ^bc^	[6.31, 7.04]	non 0	0.00023
^2^ ∑SFA	87.10	[83.9, 90.4]	89.90	[86.6, 93.1]	87.40	[84.2, 90.6]	90.50	[87.3, 93.7]	90.80	[87.6, 94.0]	non 0	0.32185
^3^ ∑MUFA	116	[112, 121]	120	[115, 124]	117	[113, 122]	118	[113, 122]	120	[115, 124]	non 0	0.71882
^4^ ∑ PUFA	54.10	[49.8, 58.4]	57.00	[52.7, 61.2]	58.00	[53.7, 62.3]	62.90	[58.6, 67.2]	57.70	[53.2, 62.0]	non 0	0.08507
∑ n-3 PUFAs	13.6 ^a^	[12.40, 14.90]	15.6 ^b^	[14.40, 16.80]	16.20 ^bc^	[15.00, 7.40]	17.90 ^c^	[16.60, 19.10]	16.10 ^bc^	[14.90, 17.30]	non 0	0.00099
∑ *n*-6 PUFA	40.20	[36.70, 43.70]	41.20	[37.70, 44.70]	41.60	[38.10, 45.10]	44.80	[41.30, 48.30]	41.30	[37.80, 44.80]	non 0	0.3988
Fat (%)	25.70	[24.9, 26.6]	26.60	[25.8, 27.5]	26.30	[25.4, 27.1]	27.10	[26.3, 27.9]	26.80	[26.0, 27.7]	non 0	0.18607

Estimated marginal means (emmeans) with 95% upper confidence (UCL) and lower confidence limits (LCL) (n = 70); emmeans within a row with ^a, b, c^ different superscript letters are significantly different at *p* < 0.05. ^1^ FS: 75 g of flaxseed + no antioxidant sources, VE8: 75 g of flaxseed + 800 mg of α-tocopherol/kg, DA8: 75 g of flaxseed + 800 mg of *D. angustifolia* extract/kg, TS8: 75 g of flaxseed + 800 mg of *T. schimperi*/kg, CD8: 75 g of flaxseed + 800 mg of *C. domestica/kg of* diet. ^2^ Sum of saturated fatty acids. ^3^ Sum of monounsaturated fatty acids. ^4^ Sum of polyunsaturated fatty acids.

**Table 3 foods-12-01819-t003:** Lipid health indices and malondialdehyde (MDA) content in egg yolks of Sasso laying hens.

^2^ Indices	Sample Type	^1^ Treatments										
		FS		VE8		DA8		TS8		CD8		*p*-Value
		Mean	95% [LCL, UCL]	Mean	95% [LCL, UCL]	Mean	95% [LCL, UCL]	Mean	95% [LCL, UCL]	Mean	95% [LCL, UCL]	
n-6/n-3 ratio	raw	3.07	[2.55, 3.59]	3.33	[2.81, 3.85]	2.61	[2.09, 3.13]	2.54	[2.02, 3.06]	2.52	[2.00, 3.04]	0.1029
	cooked	4.01	[3.77, 4.25]	3.69	[3.45, 3.93]	3.59	[3.35, 3.83]	3.55	[3.31, 3.79]	3.59	[3.35, 3.83]	0.05681
AI	raw	0.53	[0.50, 0.54]	0.52	[0.49, 0.53]	0.52	[0.50, 0.54]	0.50	[0.48, 0.52]	0.52	[0.49, 0.53]	0.3956
	cooked	0.52	[0.50, 0.53]	0.52	[0.50, 0.52]	0.51	[0.49, 0.52]	0.51	[0.49, 0.52]	0.52	[0.50, 0.53]	0.4562
TI	raw	0.75	[0.71, 0.79]	0.74	[0.69, 0.77]	0.72	[0.68, 0.76]	0.68	[0.63, 0.71]	0.70	[0.66, 0.74]	0.1104
	cooked	0.72 ^a^	[0.69, 0.75]	0.70 ^ab^	[0.67, 0.73]	0.68 ^bc^	[0.64, 0.70]	0.66 ^bc^	[0.63, 0.69]	0.70 ^abc^	[0.66, 0.72]	0.02659
h/H ratio	raw	2.42	[2.31, 2.53]	2.43	[2.32, 2.54]	2.38	[2.27, 2.49]	2.56	[2.45, 2.67]	2.46	[2.35, 2.57]	0.2229
	cooked	2.41	[2.33, 2.50]	2.44	[2.35, 2.52]	2.50	[2.41, 2.59]	2.46	[2.37, 2.55]	2.46	[2.38, 2.55]	0.6718
NVI	raw	0.75	[0.72, 0.77]	0.72	[0.70, 0.74]	0.73	[0.70, 0.75]	0.71	[0.69, 0.73]	0.73	[0.70, 0.74]	0.2179
	cooked	1.97	[1.89, 2.04]	1.97	[1.90, 2.05]	2.00	[1.92, 2.07]	1.91	[1.84, 1.99]	2.00	[1.93, 2.08]	0.417
HFA	raw	64.7	[61.9, 67.5]	66.0	[63.2, 68.8]	67.2	[64.3, 70.0]	64.8	[62.0, 67.6]	66.3	[63.5, 69.2]	0.6883
	cooked	64.7	[61.9, 67.5]	66.4	[63.7, 69.3]	64.5	[61.7, 67.3]	67.3	[64.4, 70.1]	66.4	[63.6, 69.2]	0.5578
S/P	raw	0.51	[0.49, 0.53]	0.50	[0.48, 0.52]	0.51	[0.49, 0.52]	0.49	[0.47, 0.50]	0.50	[0.48, 0.52]	0.4107
	cooked	0.51	[0.49, 0.52]	0.50	[0.49, 0.51]	0.49	[0.48, 0.50]	0.49	[0.48, 0.50]	0.51	[0.49, 0.51]	0.4324
MDA (µg/g)	raw	4.44 ^a^	[3.16, 5.72]	4.20 ^a^	[2.89, 5.51]	3.11 ^ab^	[1.80, 4.42]	1.46 ^b^	[0.16, 2.75]	3.87 ^ab^	[2.57, 5.16]	0.01476

Estimated marginal means (emmeans) with 95% upper confidence (UCL) and lower confidence limits (LCL) (n = 70); emmeans within a row with ^a, b, c^ different superscript letters are significantly different at *p* < 0.05. ^1^ FS: 75 g of flaxseed + no antioxidant sources, VE8: 75 g of flaxseed + 800 mg of α-tocopherol/kg, DA8: 75 g of flaxseed + 800 mg of *D. angustifolia* extract/kg, TS8: 75 g of flaxseed + 800 mg of *T. schimperi*/kg, CD8: 75 g of flaxseed + 800 mg of *C. domestica/kg of* diet. ^2^ AI: atherogenic indices, TI: thrombogenic indices, ratio h/H: ratio of hypo/hyper cholestrolemic fatty acids, NVI: nutritional value indices, DFA: desirable fatty acid indices, HFA: hypercholestrolemic fatty acids, s/p: saturation indices.

## Data Availability

Data are contained within the article or the Appendix A.

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
