# Peer review of "Yolk Fatty Acid Content, Lipid Health Indices, and Oxidative Stability in Eggs of Slow-Growing Sasso Chickens Fed on Flaxseed Supplemented with Plant Polyphenol Extracts"

_foods, 2023, doi:10.3390/foods12091819_

Round 1

Reviewer 1 Report

“Yolk fatty acid content, lipid health indices and oxidative stability in eggs of slow-growing Sasso chickens fed on flaxseed supplemented with plant polyphenol extracts” is the study about the effect of feeds containing flaxseed and plant polyphenols on the yolk fatty acid content and other properties of Sasso chicken eggs. The authors did multiple approaches to reach to the conclusion along with an appropriate discussion of the results. However, following issues need to be addressed before it is accepted for publication in Foods:

1. Abstract: needs thorough revision to make clear about the message the authors want to convey. Also, in p < 0.05, p should be italic. More importantly, the use of “p > 0.05” needs proper revision as per the significance condition.

2. Line 78-82: For these three medicinal plants, please give the common name and the full scientific name within a parenthesis.

3. Line 91-95: “Previously, we……… supplementation [40]” Please revise this sentence to make the message clear to the reader.

4. Line 110: Please correct as: “……and analysis were followed according to our recent publication [40].”

5. Line 115-117: Be consistent in using comma (,) and semi-colon (;).

6. Line 123-126: Please complete the sentence using an appropriate verb.

7. Line 130-133: Please complete the sentence using an appropriate verb. Same comment for line 133-140. Whole manuscript needs thorough revision of English language and grammar, especially use of a complete sentence for expression; use of punctuations as appropriate etc.

8. Line 151: “….. prepared according to Nimalaratne and coworkers [42].” Please use such way of expression; instead of writing directly the reference number (throughout the manuscript!).

9. Line 153: “…… and boiled for 10 min at 76 oC.” Water may not be boiled at 76 oC. Please correct appropriately. OR why you chose this temperature (76 oC)?

10. Line 174: “a vacuum centrifuge at 438 g/min, 30 oC, 30 min.” ?

11. Throughout the manuscript: milliliter – mL; hour – h; second – s.

12. Line 226: please correct “Where:” as “where,”

13. Line 295: “PUFA n-3” or “n-3 PUFA”? Please consistent in using a phrase throughout the manuscript.

Reviewer 2 Report

The research conducted is relevant since it suggests new sources of antioxidant compounds to prevent lipid oxidation in foods with high content of polyunsaturated fatty acids. Furthermore, the study is all the more interesting as it broadens the knowledge about local plants traditionally grown in Ethiopia.

However, some issues have to be tackled before the manuscript can be published:

- Since the antioxidant properties of the plant extracts are attributed to the plant content in phenolic compounds, I find essential to provide in this study data about concentrations in polyphenols in these plants.  

- In the "experimental design" section, it is not clear in which experimental facilities was the trial carried out.

- The word "chickens" should be avoided and replaced by "laying hens". 

- Instead of "hr", "hours" should be fully written all throughout the manuscript.

- I don't think essential to specify that birds are "slow-growing" since the study deals with egg production and not with meat production.

- L30: if (p>0.05), how can you conclude that plant extracts decreased MDA content in egg yolks?

-L30-32: conclusion in the abstract should be rephrased. It is not correctly written.

- L46: "remains" instead of "remained"

- L49: "source" instead of "component"

-L54: "laying hens" instead of "layer chicken"

-L57: "from flaxseed into EPA and DHA" and "deposition in eggs remains scarce as compared with the quantity of...."

-L59: "and to increase the level of lipid peroxidation"

-L63: "added ALA into EPA and DHA"

-L66: "plants as natural antioxidants better than synthetic antioxidant compounds"

-L67: "plants have been considered" instead of "plants having been considered"

-L75: the study of Romero et al. (2022) "Productive performance, egg quality and yolk lipid oxidation in laying hens fed diets including grape pomace or grape extract" should be cited in this paragraph

-L76: "multiethnic and diverse cultures" could be shortened as simply "diverse cultures"

-L79: rephrase the following sentences. It could be written as follows: "Furthermore, Thymus schimperi (T. schimperi), also with a high phenolic content and thereby strong antioxidant capacity, has been used to treat diabetic patients [33,34,35]"

-L81: "the" can be deleted before "traditional Ethiopian kitchen"

-L82: "which" does not need to be written in italics

-L83: "is" should be inserted in "and is an efficient antioxidant"

-L85: text could be rephrased this way: "However, there is a dearth of studies dealing with slow-growing chickens [38,39,40]"

-L88: the acronym "PUAFs" is not correct. It should be written as "PUFAs".

-L88: It does not seem correct to state that polyphenolic extracts of medicinal plants can enhance n-3 PUFAs content. This effect is due to the dietary inclusion of flaxseed.

-L89: "flaxseed as lipid source"

-L89-90: "plan polyphenol extracts as antioxidants"

-L92: Specify whether Sasso T451A are growing chickens or laying hens.

-L93: "in raw and cooked" eggs or meat? It is not clear.

-L99 (at the end): "Sasso chickens".

-L109-110: "Study details including diet composition and analysis are provided in a previous publication [40]"

-L111: "... with plant extracts and vitamin E (α-tocopherol acetate) being incorporated into a small amount of wheat bran before their mixing with the main ingredients..."

-L141: some information pertaining to the type of compounds present in the plant extracts (e.g. thymol, eugenol, carvacrol) and to the polyphenols concentrations in the extracts included in the diets should be provided here. This constitutes essential information in order to characterise the plants tested in this study. Furthermore, this information is needed to understand the results obtained.

-L149: a comma should be inserted after "Then"

-L151: "prepared according to Nimalaratne et al. [42]"

-L195: it is not clear whether MDA concentration was measured in freshly collected eggs or in stored eggs. If eggs had been stored before the MDA determination, please specify storage conditions and for how long eggs had been stored prior to this analysis.

-L435-438: results about oxidative stability in egg yolks obtained in the present study should be compared with the results shown in the work of Romero et al. (2022) "Productive performance, egg quality and yolk lipid oxidation in laying hens fed diets including grape pomace or grape extract". It would be very interesting to compare your results with those obtained when including grape byproducts in the diets of laying hens.

-L439-440: check whether it is correct that p > 0.05. If p > 0.05, you cannot state that there was an effect of including vitamin E in the diet on egg yolk MDA concentration.

-L451-464: you should not use abbreviations like TS8 or DA8 in the conclusions. Moreover, you have to be more specific and accurate when writing the conclusions. For instance, you have to specify which plant extracts have been tested in this research work. After writing "a moderate dose of flaxseed", please specify between brackets the dose.

-L453: instead of "an important approach", it could be said "a useful approach"

-L454-455: "while mantaining favourable atherogenic and thrombogenic indices and a suitable n-6/n-3 PUFA ratio for consumer health"

-L459: "but was not as efficient as plant polyphenol extracts". At the beginning of the conclusions, it has to be specified which plant extracts you are evaluating.

-L461: "indigenous plants"

-L462-464: the issues suggested as future research are very pertinent.

-L489: "Data are contained within...." instead of "Data is contained within...."

Reviewer 3 Report

Dear,

The paper is related to yolk fatty acid content, lipid health indices and oxidative stability in eggs of slow-growing Sasso chickens fed on flaxseed supplemented with plant polyphenol extracts. This topic is not new and there have been lot of papers published so far.

Previously, the same authors showed that feeding Sasso chicken on flaxseed together with polyphenol extract of C. domestica in raw and cooked and T. schimperi in raw breast muscle significantly increased the content of C22: 6n-3 (DHA) and C20:5n-3 (EPA) contents compared to sole flaxseed supplementation. This paper is some sort of extension of their previous work. In my opinion authors should emphasised novelty in the present work. What is new? This should be added in Introduction part.

Also, cooked and raw egg yolks were analysed, but that is not mentioned in Abstract

Novelty shod be clarified. There are also some additional points:

Line 88: Correct “PUAF” to “PUFA”

Provide number of analysed samples (n) in Tables. Tables should be self-explanatory. The reader should be able to understand the contents of the table without referring to the text.

Round 2

Reviewer 1 Report

“Yolk fatty acid content, lipid health indices and oxidative stability in eggs of slow-growing Sasso chickens fed on flaxseed supplemented with plant polyphenol extracts” On analyzing the revised version of the manuscript against the issues raised, following previous points needs further revision:

Comment No. 1: Instead of the authors claim, no correction was seen in terms of italicized p-value (p).

Comment No. 9: Your claim about boiling point of water boils at 2355 meters altitude as 76 oC is not valid. Please see the following information:

Therefore, I suggest you to use other terminologies than “boiling”.

Comment No. 10: Vacuum centrifuge or vacuum evaporator? Also, the unit 438 g, 30 oC, 30 min? Please use a proper terminology and proper unit.

Reviewer 3 Report

Dear,

In my opinion, the manuscript has been carefully revised according to the suggestions and comments of the reviewers. The quality of the manuscript has been significantly improved. I suggest that the manuscript should be accepted. 

Author Response

We appreciate your insightful comments and suggestions, which have improved the manuscript's quality.